# Synthetic lethality between the cohesin subunits *STAG1* and *STAG2* in diverse cancer contexts

Petra van der Lelij[1†], Simone Lieb[2†], Julian Jude[1], Gordana Wutz[1], Catarina P Santos[3,4], Katrina Falkenberg[1], Andreas Schlattl[2], Jozef Ban[5], Raphaela Schwentner[5], Thomas Hoffmann[1], Heinrich Kovar[5,6], Francisco X Real[3,4,7], Todd Waldman[8], Mark A Pearson[2], Norbert Kraut[2], Jan-Michael Peters[1], Johannes Zuber[1], Mark Petronczki[2]*

[1]Research Institute of Molecular Pathology, Vienna Biocenter, Vienna, Austria; [2]Boehringer Ingelheim RCV GmbH & Co KG, Vienna, Austria; [3]Spanish National Cancer Research Centre, Madrid, Spain; [4]Centro de Investigación Biomédica en Red de Cáncer, Madrid, Spain; [5]Children's Cancer Research Institute, Vienna, Austria; [6]Department for Pediatrics, Medical University of Vienna, Vienna, Austria; [7]Department de Ciències Experimentals I de la Salut, Universitat Pompeu Fabra, Barcelona, Spain; [8]Lombardi Comprehensive Cancer Center, Georgetown University School of Medicine, Washington DC, United States

**\*For correspondence:** mark_paul. petronczki@boehringer-ingelheim. com

[†]These authors contributed equally to this work

**Abstract** Recent genome analyses have identified recurrent mutations in the cohesin complex in a wide range of human cancers. Here we demonstrate that the most frequently mutated subunit of the cohesin complex, *STAG2*, displays a strong synthetic lethal interaction with its paralog *STAG1*. Mechanistically, STAG1 loss abrogates sister chromatid cohesion in *STAG2* mutated but not in wild-type cells leading to mitotic catastrophe, defective cell division and apoptosis. STAG1 inactivation inhibits the proliferation of STAG2 mutated but not wild-type bladder cancer and Ewing sarcoma cell lines. Restoration of STAG2 expression in a mutated bladder cancer model alleviates the dependency on STAG1. Thus, STAG1 and STAG2 support sister chromatid cohesion to redundantly ensure cell survival. STAG1 represents a vulnerability of cancer cells carrying mutations in the major emerging tumor suppressor *STAG2* across different cancer contexts. Exploiting synthetic lethal interactions to target recurrent cohesin mutations in cancer, e.g. by inhibiting STAG1, holds the promise for the development of selective therapeutics.

## Introduction

Cohesin is a highly conserved ring-shaped protein complex that is thought to topologically embrace chromatid fibers (*Peters and Nishiyama, 2012*), which is essential for sister chromatid cohesion and chromosome segregation in eukaryotes. In addition, cohesin participates in DNA repair, genome organization and gene expression (*Losada, 2014*). The cohesin subunits SMC1, SMC3 and RAD21 (also called SCC1) comprise the core ring of the complex. A fourth universally conserved subunit, a HEAT repeat protein of the Scc3/STAG family, peripherally associates with the core cohesin ring by binding to RAD21 (*Tóth et al., 1999*), and is required for the dynamic association of cohesin with chromatin (*Hu et al., 2011*; *Murayama and Uhlmann, 2014*). Human somatic cells express two paralogs of this protein, called STAG1 and STAG2 (*Losada et al., 2000*; *Sumara et al., 2000*).

**eLife digest** A big challenge for cancer research is to find drugs and treatments that kill cancer cells without harming the other cells of a patient. Cancer cells contain genetic mutations that cause them to grow and divide more rapidly than healthy cells. About half a million cancer patients worldwide have tumors that feature mutations to the gene that produces a protein called STAG2, a component of a large protein ring called cohesin. These mutations are particularly common in bladder cancers and Ewing sarcoma, a childhood bone cancer.

The cohesin ring holds together duplicated chromosomes during cell division, establishing the iconic X-shape of chromosomes in dividing cells. It is not clear exactly how mutations that affect STAG2 make cancer more likely to develop. However, it is possible that these cancer-specific mutations make cancer cells vulnerable in ways that healthy cells are not.

Using a genetic screening approach, van der Lelij, Lieb et al. searched for genes whose inactivation would harm only those cells that have mutant STAG2 proteins. This search found that one such gene encodes a protein called STAG1, a close relative of STAG2. Reducing the amount of STAG1 protein in cells with mutant forms for STAG2 caused these cells to start dying, whereas healthy cells were unaffected.

Van der Lelij, Lieb et al. then conducted biochemical and cell biological experiments on bladder cancer and Ewing sarcoma cells to show that the cells need at least one of STAG1 or STAG2 to hold replicated chromosomes together. Without either protein, the X-shape of the chromosomes was lost and the cells died when they tried to divide.

Thus, human cells can survive without STAG1 or STAG2 but not without both, a concept known as synthetic lethality. More research is now needed to identify how the STAG1 protein could be prevented from working. This knowledge could ultimately be used to develop drugs that would kill off only those cancer cells that have mutations that affect STAG2.

Recent cancer genome studies identified recurrent mutations in cohesin subunits and regulators in approximately 7.3% of all human cancers (*Lawrence et al., 2014*; *Leiserson et al., 2015*; *Solomon et al., 2011*). *STAG2*, the most frequently mutated cohesin subunit, emerges as one of only 12 genes that are significantly mutated in four or more major human malignancies (*Lawrence et al., 2014*). *STAG2* mutations have been reported in ~6% of acute myeloid leukemias and myelodysplastic syndromes (*Kon et al., 2013*; *Thota et al., 2014*; *Walter et al., 2012*), 15–22% of Ewing's sarcomas (*Brohl et al., 2014*; *Crompton et al., 2014*; *Tirode et al., 2014*), and in up to 26% of bladder cancers of various stages and grades (*Balbás-Martínez et al., 2013*; *Guo et al., 2013*; *Solomon et al., 2013*; *Taylor et al., 2014*). The deleterious nature of most *STAG2* mutations strongly suggests that the gene represents a new tumor suppressor (*Hill et al., 2016*). *STAG2* mutations were initially thought to promote tumorigenesis due to defects in sister chromatid cohesin leading to genome instability (*Barber et al., 2008*; *Solomon et al., 2011*). However, the vast majority of cohesin-mutated cancers are euploid (*Balbás-Martínez et al., 2013*; *Kon et al., 2013*), indicating that cohesin mutations may promote tumorigenesis through altering different cohesin functions such as genome organization and transcriptional regulation (*Galeev et al., 2016*; *Mazumdar et al., 2015*; *Mullenders et al., 2015*; *Viny et al., 2015*). Regardless of the mechanisms driving cohesin mutant tumors, the recent success of poly(ADP-ribose) polymerase inhibitors in the treatment of *BRCA*-mutated ovarian and prostate cancer demonstrates that exploiting tumor suppressor loss by applying the concept of synthetic lethality in defined patient populations can impact clinical cancer care (*Castro et al., 2016*; *Kim et al., 2015*; *Mirza et al., 2016*; *Oza et al., 2015*). The estimated half a million individuals with *STAG2*-mutant malignancies would greatly profit from exploring specific dependencies of these cancers.

## Results

We hypothesized that STAG2 loss could alter the properties and function of the cohesin complex leading to unique vulnerabilities of *STAG2* mutated cells. To identify factors whose inactivation would be synthetic lethal with loss of STAG2 function, we first used CRISPR/Cas9 to inactivate

STAG2 in near-diploid, chromosomally stable HCT 116 colon carcinoma cells (*Figure 1A*). Two clones, STAG2- 505c1 and 502c4, harboring deleterious mutations in *STAG2* and lacking detectable STAG2 protein expression were selected for analyses (*Figure 1—figure supplement 1* and *Supplementary file 1*). The isogenic parental and STAG2- HCT 116 cells were transfected with short-interfering RNA (siRNA) duplexes targeting 25 known cohesin subunits and regulators. After normalization to the non-target control siRNA (NTC), the effects of siRNA duplexes targeting individual genes were compared in parental and STAG2- cells. Depletion of the known essential cohesin regulator SGOL1 had a detrimental impact on viability of both parental and STAG2- cells. Remarkably, depletion of STAG1 strongly decreased cell viability in STAG2- cells, while being tolerated by the isogenic parental cells (*Figure 1B*). The pronounced selective effect of STAG1 depletion on STAG2- cells was confirmed in individual transfection experiments and colony formation assays (*Figure 1C,D,E*). Expression of an siRNA-resistant STAG1 transgene alleviated the anti-proliferative effect of STAG1 but not of SGOL1 siRNA duplexes in STAG2- HCT 116 cells demonstrating the specificity of the siRNA treatment (*Figure 1—figure supplement 2*). Double depletion of STAG1 and STAG2 by siRNA in parental cells confirmed their synthetic lethal interaction (*Figure 1—figure supplement 3*). Co-depletion of p53 and STAG1 indicated that the dependency of STAG2- cells on STAG1 was independent of p53 (*Figure 1—figure supplement 4*). In contrast to the loss of essential cohesin subunits or regulators, depletion of STAG1 had no effect on cell viability in non-transformed telomerase-immortalized human retinal pigment epithelial cells (hTERT RPE-1) (*Figure 1—figure supplement 5*). This result is supported by a large-scale genetic loss-of-function study that found that neither *STAG1* nor *STAG2* is essential for the proliferation of hTERT-RPE1 cells (*Hart et al., 2015*). To corroborate our genetic interaction findings using an independent strategy, we introduced Cas9 into parental and STAG2- HCT 116 cells as well as KBM-7 leukemia cells for competition assays (*Figure 1F* and *Figure 1—figure supplement 1*). Transduction of lentiviruses co-expressing mCherry and single guide RNAs (sgRNAs) targeting essential cohesin subunit genes, such as *RAD21* and *SMC3*, resulted in the rapid loss of the infected and mCherry-positive cells from the population irrespective of *STAG2* genotype (*Figure 1F*). In striking contrast, transduction with sgRNAs targeting *STAG1* caused the depletion of STAG2- HCT 116 and KBM-7 cells but not of their parental *STAG2* proficient counterparts (*Figure 1F*). Collectively, these experiments identify STAG1 as a vulnerability of *STAG2* mutated cells in engineered solid cancer and leukemia models. STAG1 inactivation has little if any impact on the viability and proliferation of *STAG2* wild-type cancer cells and non-transformed cells, but is essential for survival in the absence of STAG2.

To elucidate the mechanistic basis for this synthetic lethal interaction, we hypothesized that the combined loss of STAG1 and STAG2, in contrast to loss of either component alone, could severely impair cell division. Chromosome alignment and segregation during mitosis rely on sister chromatid cohesion, the central function of the cohesin complex (*Peters and Nishiyama, 2012*). Depletion of STAG1 resulted in an increase in the mitotic index and a prolongation of the duration of mitosis in STAG2- but not wild-type cells (*Figure 2A* and *Figure 2—figure supplement 1*). Immunofluorescence microscopy revealed a failure to align chromosomes at the metaphase plate upon STAG1 loss in STAG2- cells (*Figure 2B*). In mitotic chromosome spread analysis *STAG2* inactivation caused a partial loss of centromeric cohesion in HCT 116 cells as previously reported (*Canudas and Smith, 2009*; *Kim et al., 2016*; *Remeseiro et al., 2012*; *Solomon et al., 2011*) (*Figure 2C*). Depletion of the essential centromeric cohesin protection factor SGOL1 resulted in a complete loss of sister chromatid cohesion in most chromosome spreads irrespective of *STAG2* genotype. In striking contrast, STAG1 depletion selectively abrogated sister chromatid cohesion in STAG2- but not parental cells (*Figure 2C*, single chromatids). The severe mitotic defects observed upon loss of STAG1 in STAG2- cells were accompanied by the emergence of aberrantly sized and shaped interphase nuclei (*Figure 2—figure supplement 2*) and by a progressive increase in apoptosis (*Figure 2D*). These results provide a mechanistic basis for the synthetic lethal interaction between *STAG1* and *STAG2. STAG1* inactivation abrogates sister chromatid cohesion exclusively in STAG2- cells resulting in catastrophic mitotic failure, abnormal cell division and apoptosis. To hold sister chromatids together, cohesin can tolerate the loss of either STAG1 or STAG2 alone but not the loss of both.

We next expanded our analysis to patient-derived *STAG2* mutations and *STAG2*-mutant cancer cell lines in order to investigate the disease relevance of the observed synthetic lethality (*Figure 3*). STAG1 depletion by siRNA abrogated both cell viability and sister chromatid cohesion in HCT 116 cell clones, in which three patient-derived deleterious mutations had been engineered into the

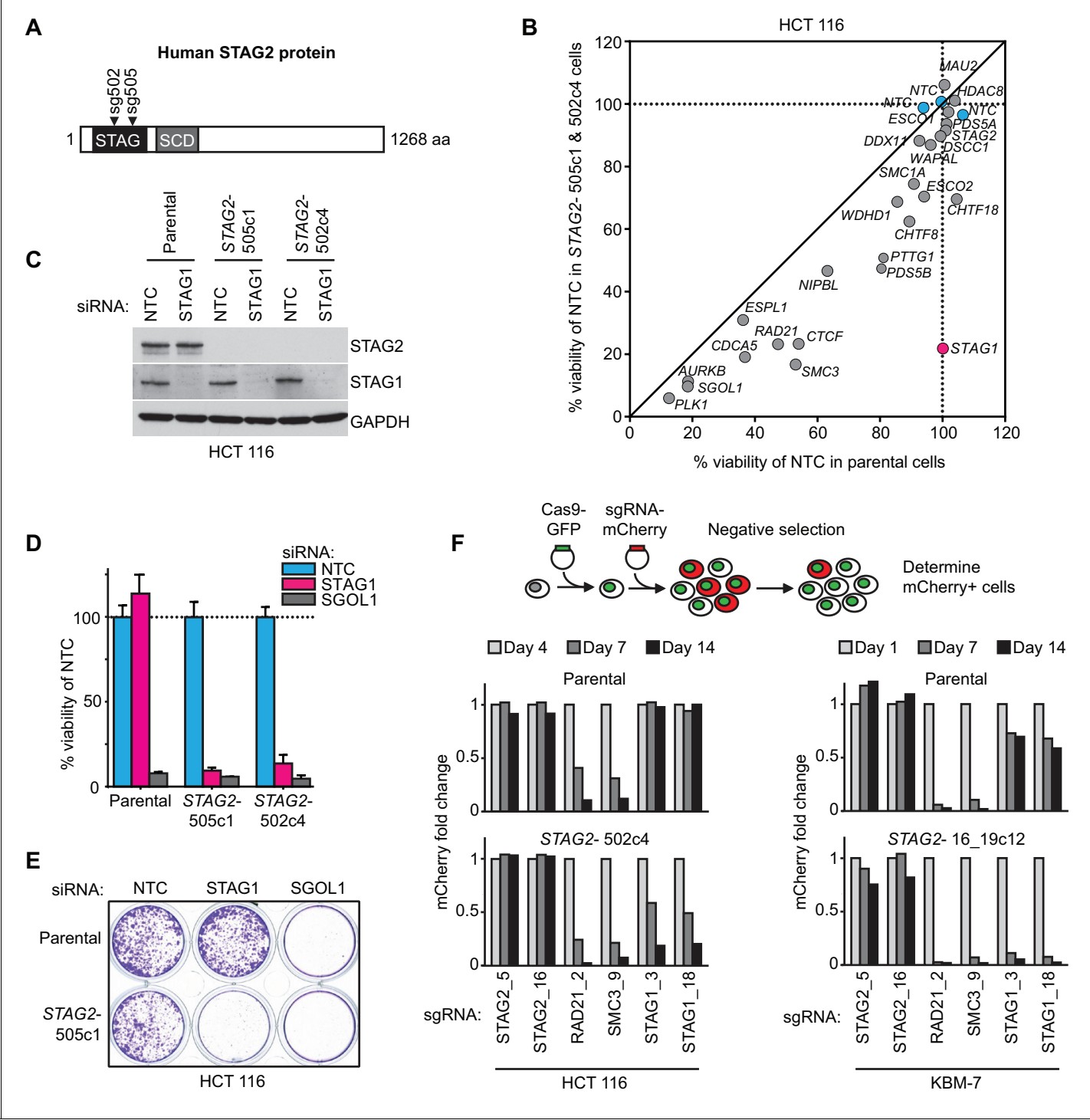

**Figure 1.** Identification of *STAG1* as a genetic vulnerability of *STAG2* mutated cells. (**A**) Engineering of an isogenic HCT 116 cell model by CRISPR/Cas9-mediated inactivation of *STAG2*. The position of the sgRNAs used to create deleterious insertion and deletion mutations in the *STAG2* coding sequence is indicated. (**B**) Parental HCT 116 cells and two *STAG2* mutant clones (*STAG2-*) were subjected to an siRNA screen. Pools of 4 siRNA duplexes targeting 25 known cohesin subunits and regulators were transfected into the three cell lines. Cell viability was measured 7 days after transfection. Following normalization to non-target control (NTC) values, the average cell viability of siRNA pools in parental HCT 116 cells and two *STAG2-* clones was plotted against each other (n = 2 or more independent experiments with 2 biological repeats each). (**C**) HCT 116 parental cells and *STAG2-* clones were transfected with the indicated siRNAs. Protein lysates were prepared 72 hr after transfection and analyzed by immunoblotting. (**D**) Cell viability was assessed 8 days after siRNA transfection using a metabolic assay (n = 4 biological repeats, error bars denote standard deviation) and (**E**) using crystal violet staining. (**F**) Cas9-GFP expressing isogenic parental and *STAG2-* HCT 116 and KBM-7 cells were transduced with a lentivirus

*Figure 1 continued on next page*

*Figure 1 continued*

encoding mCherry and sgRNAs targeting the indicated genes. The percentage of mCherry-positive cells was determined over time by flow cytometry and normalized to the fraction of mCherry-positive cells at the first measurement and sequentially to a control sgRNA (n = 1 experimental replicate).

The following figure supplements are available for figure 1:

**Figure supplement 1.** Characterization of CRISPR/Cas9-generated *STAG2* mutated clones used in this study.

**Figure supplement 2.** Rescue of the synthetic lethal interaction between *STAG1* and *STAG2* by expression of an siRNA-resistant FLAG-STAG1 transgene.

**Figure supplement 3.** Double depletion experiment confirms *STAG1-STAG2* synthetic lethality.

**Figure supplement 4.** Double depletion experiments indicate that the *STAG1-STAG2* genetic interaction is independent of p53.

**Figure supplement 5.** Depletion of STAG1 does not reduce viability in hTERT RPE-1 cells.

*STAG2* locus (*Kim et al., 2016*), but not in parental HCT 116 cells (*Figure 3—figure supplement 1*). Among solid human cancers, *STAG2* mutational inactivation is most prevalent in urothelial bladder cancer and Ewing sarcoma. Therefore, we assembled a panel of 16 bladder cancer cell lines: 11 STAG2-positive, three with deleterious *STAG2* mutations (UM-UC-3, UM-UC-6 and VM-CUB-3), one in which *STAG2* was inactivated by CRISPR/Cas9 (UM-UC-5 *STAG2*- 505c6) (*Figure 3—figure supplement 2*), and two with no detectable STAG2 expression (LGWO1 and MGH-U3) (*Supplementary file 2*) (*Balbás-Martínez et al., 2013*; *Solomon et al., 2013*). The STAG2 protein expression status in the panel of bladder cancer cell lines was confirmed using immunoblotting (*Figure 3A*). siRNA experiments revealed that STAG2 status represented a predictive marker for the sensitivity to STAG1 depletion across the bladder cancer cell line panel. Whereas all cell lines were highly sensitive to depletion of the key mitotic kinase PLK1, STAG1 siRNA reduced cell viability in *STAG2*-negative bladder cancer cells but had little or no effect on STAG2-positive bladder cancer cell lines (*Figure 3B*). STAG1 depletion prevented colony formation and abolished sister chromatid cohesion selectively in *STAG2* mutated UM-UC-3 (F983fs) but not in *STAG2* wild-type UM-UC-5 bladder cancer cells (*Figure 3C,D*). In contrast, SGOL1 depletion abrogated cell growth and cohesion in both cell lines. Consistent with the results obtained in bladder cancer cells, *STAG2* mutation status was also linked to STAG1 dependency in a panel of four Ewing sarcoma cell lines (*Figure 3E, F* and *Supplementary file 2*) (*Solomon et al., 2011*; *Tirode et al., 2014*). Lentiviral transduction of a FLAG-STAG2 transgene into *STAG2* mutated UM-UC-3 bladder cancer cells resulted in the restoration of STAG2 expression, nuclear localization of the transgenic protein and its incorporation into the cohesin complex (*Figure 3—figure supplement 3*). Crucially, restoration of STAG2 expression alleviated the STAG1 dependency of UM-UC-3 cells providing a causal link between STAG2 loss and STAG1 dependency (*Figure 3G*). These results demonstrate that the synthetic lethal interaction between STAG1 and STAG2 that we discovered in isogenic cell pairs is recapitulated in disease-relevant bladder cancer and Ewing sarcoma cell models.

## Discussion

Here we identify *STAG1* as a strong genetic vulnerability of cells lacking the major emerging tumor suppressor STAG2 (*Figure 4*). The paralog dependency between STAG1 and STAG2 has recently also been reported in an independent study (*Benedetti et al., 2017*). We show that the synthetic lethal interaction between *STAG1* and *STAG2* is observed in isogenic HCT 116 and KBM-7 cells as well as in bladder cancer and Ewing sarcoma cell lines. Thus, the genetic interaction between *STAG* paralogs is conserved in three major human malignancies: carcinoma, leukemia and sarcoma. Importantly, the finding that cancer cells harboring deleterious *STAG2* mutations remain exquisitely dependent on STAG1 demonstrates that this genetic vulnerability is maintained throughout the process of carcinogenesis and not bypassed by adaptive processes, such as the transcriptional activation of the germline-specific paralog *STAG3* (*Pezzi et al., 2000*; *Prieto et al., 2001*). Furthermore,

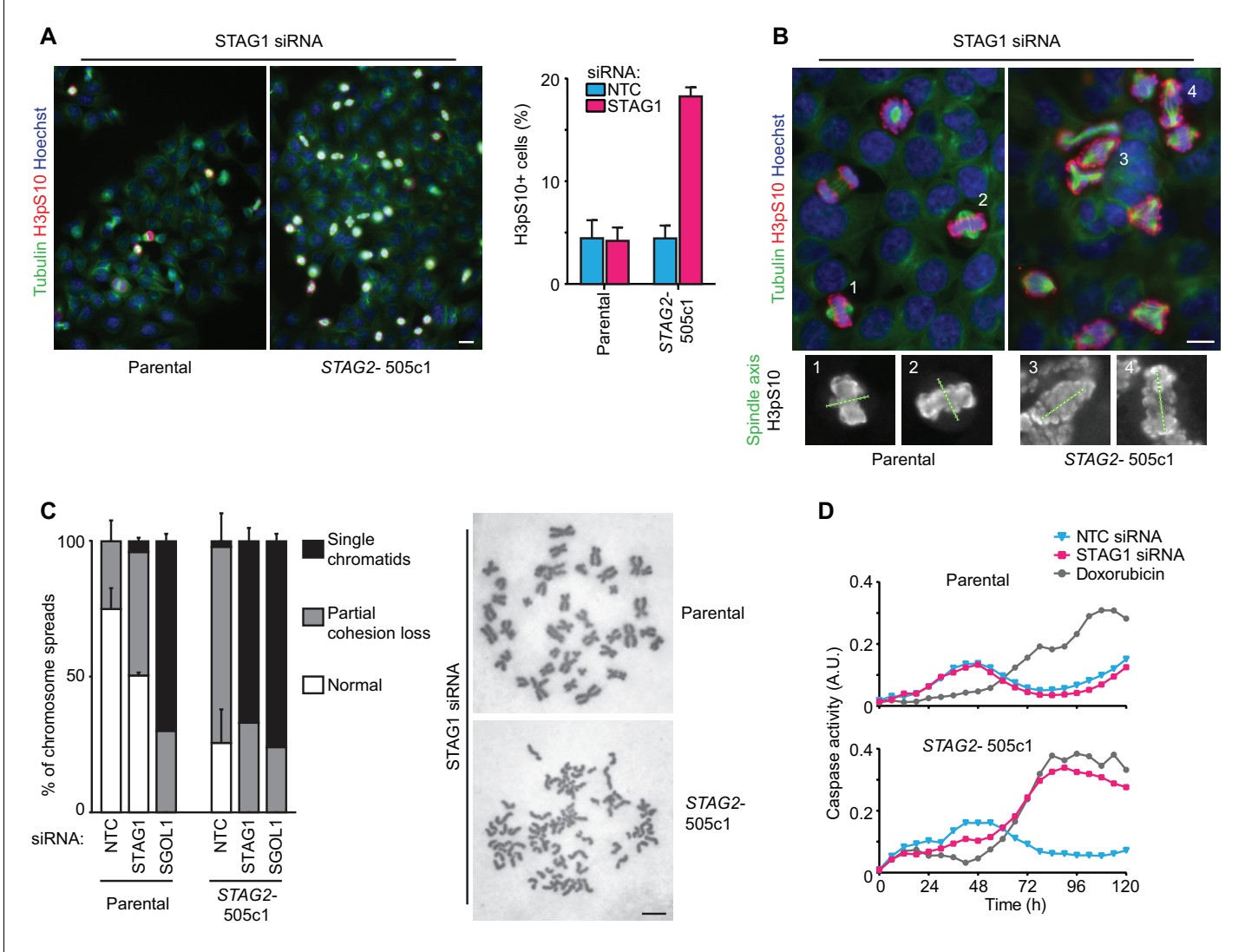

**Figure 2.** Loss of STAG1 function causes severe mitotic defects, abrogates sister chromatid cohesion and triggers apoptosis in *STAG2*- but not parental HCT 116 cells. (A) Parental and *STAG2*- 505c1 HCT 116 were transfected with NTC and STAG1 siRNA duplexes. Immunofluorescence analysis was performed 72 hr after transfection to determine the mitotic index by scoring the fraction of histone H3 phosphoSer10-positive (H3pS10+) cells (n ≥ 1323 cells, error bars denote standard deviation of three independent experiments), and (B) to investigate mitotic spindle geometry and chromosome alignment. Cropped and magnified examples of chromosome alignment are shown in (B). Scale bars, 20 µm. (C) Giemsa-stained mitotic chromosome spreads were prepared 48 hr after transfection of parental and *STAG2*- HCT 116 cells with the indicated siRNA duplexes. The status of sister chromatid cohesion of individual metaphase spreads was categorized into normal, partial loss of cohesion or single chromatid phenotypes (n = 100 spreads, error bars denote standard deviation of two independently analyzed slides). Scale bar, 10 µm. (D) Caspase activity was tracked over time using a live-cell caspase 3/7 substrate cleavage assay in parental and *STAG2*- 505c1 HCT 116 cells after transfection with the indicated siRNAs or after treatment with 0.3 µM doxorubicin at t = 0 hr (n = 2 independent experiments with 4 biological repeats for NTC and STAG1 siRNA each and 1 biological repeat for doxorubicin each).

The following figure supplements are available for figure 2:

**Figure supplement 1.** The depletion of STAG1 prolongs mitosis in *STAG2* mutated but not parental HCT 116 cells.

**Figure supplement 2.** Aberrant nuclear morphology in *STAG2*-mutated but not parental HCT 116 cells upon depletion of STAG1.

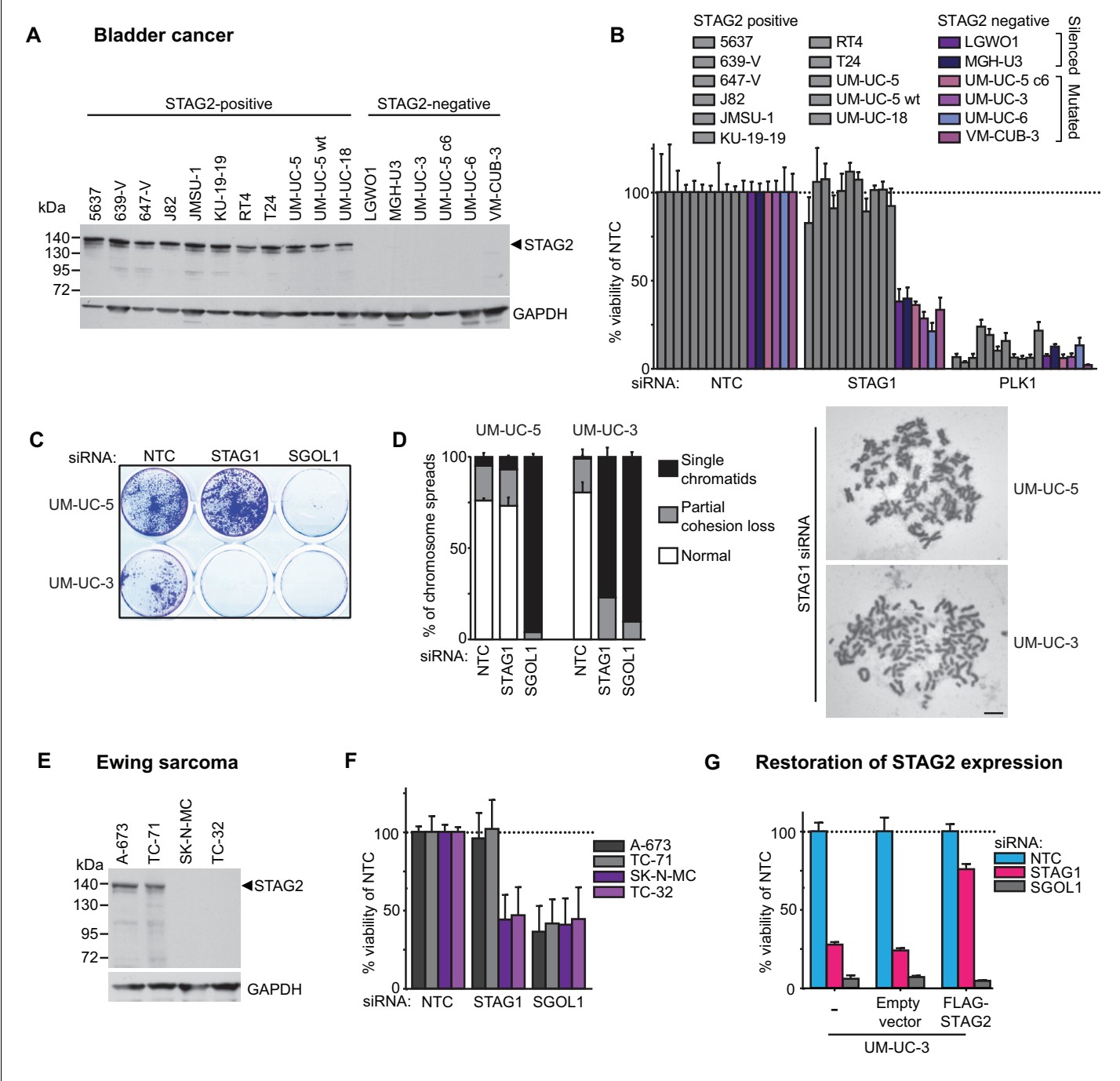

**Figure 3.** The synthetic lethal interaction between *STAG1* and *STAG2* is manifested in disease-relevant bladder cancer and Ewing sarcoma cell lines. (A) The indicated bladder cancer cell lines were analyzed for STAG2 expression by immunoblotting. (B) The indicated bladder cancer cell lines were transfected with NTC, STAG1 and PLK1 siRNA duplexes. Viability was determined 7 or 10 days after transfection and normalized to the viability of NTC siRNA transfected cells (n = 2 independent experiments with 5 biological repeats each, error bars denote standard deviation). (C) *STAG2* wild-type UM-UC-5 and *STAG2* mutated UM-UC-3 cells were transfected with NTC, STAG1 and SGOL1 siRNA duplexes. Colony formation was analyzed 7 days after transfection by crystal violet staining. (D) 72 hr after siRNA transfection into UM-UC-5 and UM-UC-3 cells, Giemsa-stained chromosome spreads were prepared and analyzed for sister chromatid cohesion phenotypes (n = 100 spreads, error bars denote standard deviation of two independently analyzed slides). (E) The indicated Ewing sarcoma cell lines were analyzed for STAG2 protein expression by immunoblotting. (F) The indicated Ewing sarcoma cell lines were transfected with NTC, STAG1 and SGOL1 siRNA duplexes. Viability was measured 6 days after transfection and normalized to the viability of NTC siRNA transfected cells (n = 3 independent experiments with 3 biological replicates each, error bars denote standard deviation). (G) *STAG2* mutated UM-UC-3 cells were transduced with a lentivirus encoding a FLAG-STAG2 transgene. Stably selected cell pools were subsequently

*Figure 3 continued on next page*

*Figure 3 continued*

transfected with NTC, STAG1 or SGOL1 siRNA duplexes. Viability was measured 7 days after transfection and normalized to the viability of NTC siRNA transfected cells (n = 4 biological replicates, error bars denote standard deviation).

The following figure supplements are available for figure 3:

**Figure supplement 1.** Patient-derived *STAG2* mutations cause STAG1 dependency in engineered isogenic HCT 116 cells.

**Figure supplement 2.** Characterization of CRISPR/Cas9-generated *STAG2* knockout in UM-UC-5 bladder cancer cell line.

**Figure supplement 3.** Restoration of STAG2 protein expression in UM-UC-3 bladder cancer cells.

expression analysis did not reveal upregulation of STAG1 protein or mRNA as a major compensatory mechanism for the loss of STAG2 function in cancer cell lines or patient tumors (*Figure 4—figure supplement 1*).

Our experiments strongly suggest that the loss of sister chromatid cohesion followed by aberrant cell division and cell death is the mechanistic basis underlying the synthetic lethality between *STAG1* and *STAG2* (*Figure 4*). Both paralogs associate with the cohesin complex in a mutually exclusive manner (*Losada et al., 2000*; *Sumara et al., 2000*). Although STAG1 and STAG2 may confer distinct functionalities to the cohesin complex (*Canudas and Smith, 2009*; *Remeseiro et al., 2012*), STAG1 and STAG2 containing complexes act redundantly to ensure sufficient sister chromatid cohesion to support cell division in human somatic cells. While the loss of one paralog is compatible with cell viability and proliferation, the loss of both paralogs abrogates cohesin's ability to hold sister chromatids

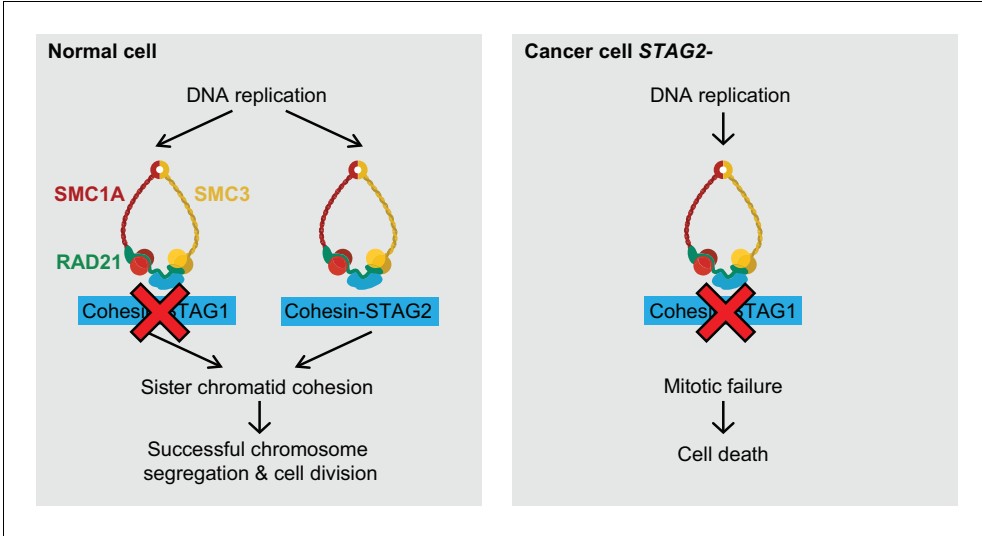

**Figure 4.** Model for the synthetic lethal interaction between *STAG1* and *STAG2*. In wild-type cells, both cohesin-STAG1 and cohesin-STAG2 complexes redundantly contribute to sister chromatid cohesion and successful cell division. Loss of STAG1 is tolerated in these cells as cohesin-STAG2 complexes alone suffice to support sister chromatid cohesion for cell division. In cancer cells in which *STAG2* is mutationally or transcriptionally inactivated, sister chromatid cohesion is now entirely dependent on cohesin-STAG1 complexes. Inactivation of STAG1 in *STAG2* mutated cells therefore results in a loss of sister chromatid cohesion followed by mitotic failure and cell death.

The following figure supplements are available for figure 4:

**Figure supplement 1.** Analysis of STAG1 protein or mRNA expression in cancer cell lines and patient tumors.

**Figure supplement 2.** Expression levels of STAG1 and STAG2 mRNA in normal human tissues.

together, which results in lethality. These mechanistic findings are consistent with previous studies that employed double siRNA depletion experiments to indicate functional redundancy between STAG1 and STAG2 in maintaining sister chromatid cohesion (*Canudas and Smith, 2009*; *Hara et al., 2014*; *Remeseiro et al., 2012*).

Since STAG1 inactivation has little or no effect on the proliferation of STAG2 proficient cells, selective targeting of STAG1 could offer a large therapeutic window. The fact that STAG2 mRNA is expressed in all normal human tissues analyzed (*Figure 4—figure supplement 2*) supports the hypothesis that selective inhibition of STAG1 function would indeed spare most non-cancerous tissues. Potential approaches for therapeutic targeting of STAG1 include the inhibition of the interaction between STAG1 and the cohesin ring subunit RAD21 (*Hara et al., 2014*) and the selective degradation of STAG1 using proteolysis-targeting chimera (PROTAC) technology (*Deshaies, 2015*). The high degree of homology between the STAG1 and its paralog STAG2 will be a key challenge to overcome. The mechanisms by which mutations in *STAG2* and other cohesin subunits drive tumorigenesis in solid and hematological tissues are not yet firmly established. Our work highlights the fact that such knowledge is not a prerequisite for the identification of selective vulnerabilities.

Both deleterious *STAG2* mutations and the loss of STAG2 expression are strong predictive biomarkers for STAG1 dependence in cell models and could be utilized for patient stratification in the future. Our work demonstrates that unique genetic dependencies of cohesin mutated cancer cells exist. Such vulnerabilities hold the promise for the development of selective treatments for patients suffering from *STAG2* mutated cancer.

## Materials and methods

### Antibodies

The following antibodies were used: C-term. pAb goat anti-STAG2 (Bethyl, Montgomery, TX, US; A300-158A and A300-159A), N-term. mAb rabbit anti-STAG2 (Cell Signaling, Danvers, MA, US; 5882), full length pAb rabbit anti-STAG2 (Cell Signaling, Danvers, MA, US; 4239), mouse anti-GAPDH (Abcam, UK; ab8245), rabbit anti-STAG1 (GeneTex, Irvine, CA, US; GTX129912), mouse anti-FLAG (Sigma-Aldrich, St. Louis, MO, US; F3165 and F1804), mouse anti-p53 (Calbiochem, San Diego, CA, US; OP43), rabbit anti-H3pS10 (Merck Millipore, Billerica, MA, US; 06570), FITC Conjugated mouse anti-Tubulin (Sigma-Aldrich, St. Louis, MO, US; F2168), rabbit anti-SGOL1 (Peters laboratory ID A975M), SMC3 (Peters laboratory ID 845), mouse anti-RAD21 (Merck Millipore, Billerica, MA, US; 05–908), rabbit anti-SMC1 (Bethyl, Montgomery, TX, US; A300-055A), and secondary rabbit (P0448), mouse (P0161) and goat (P0160) anti-IgG-HRP (all Dako, Denmark).

### CRISPR/Cas9 and cDNA transgene vectors

sgRNA sequences used in this study are listed in *Supplementary file 1*. The following lentiviral vectors were used to introduce mutations in *STAG2* in HCT 116 and UM-UC-5 cells: CRISPR STAG2 Hs0000077505_U6gRNA-Cas9-2A-GFP and CRISPR STAG2 Hs0000077502_U6gRNA-Cas9-2A-GFP (Sigma-Aldrich, St. Louis, MO, US). CRISPR sgRNA STAG2_16 and sgRNA STAG2_19 were co-expressed from U6gRNA 16-U6gRNA 19-EF1αs-Thy1.1-P2A-neo to introduce mutations in *STAG2* in KBM-7 cells. HCT 116 stably expressing Cas9-GFP were obtained by lentiviral transduction with pLentiCRISPR-EF1αs-Cas9-P2A-GFP-PGK-puro. KBM-7 infected with dox-inducible Cas9 were obtained by sequential retroviral transduction with pWPXLd-EF1A-rtTA3-IRES-EcoRec-PGK-Puro and pSIN-TRE3G-Cas9-P2A-GFP PGK-Blast. pLVX-3xFLAG-STAG1r-IRES-Puro and pLVX-3xFLAG-STAG2r-IRES-Puro lentiviral vectors for siRNA-resistant transgene expression were generated by gene synthesis (GenScript, China) based on the STAG1 cDNA sequence NCBI NM_005862.2 and STAG2 cDNA sequence NCBI NM_001042749.2 followed by cloning into the parental pLVX-IRES-Puro vector (Clontech, Mountain View, CA, US). Silent nucleotide changes were introduced into the *STAG1* and *STAG2* coding sequences within the siRNA target sequences to render the transgenes siRNA-resistant. For competition assays, U6-sgRNA-EF1αs-mCherry-P2A-neo lentiviral vectors were used.

## Cell culture and lentiviral transduction

HCT 116 cells were cultured in McCoy's 5A w/glutamax medium supplemented with 10% fetal calf serum (FCS), KBM-7 cells were cultured in IMDM medium supplemented with 10% FCS, sodium butyrate, L-glutamine and antibiotics (all Invitrogen, Waltham, MA, US). hTERT RPE-1 cells were cultured in DMEM:F12 (ATCC: 30–2006)+10% FCS+0,01 mg/ml hygromycin B. Bladder cancer cell lines UM-UC-5, UM-UC-6, UM-UC-18, LGWO1, and MGHU-3 were cultured in DMEM +10% FCS w/ NEAA, Glutamax and NaPyruvat, 5637, 639 V, 647 V, J82, JMSU-1, KU-19–19, RT4 T24, UM-UC-3 and VM-CUB-3 were cultured according to ATCC instructions (ATCC, Manassas, VA, USA). All Ewing sarcoma cell lines were cultured in RPMI +10% FCS. Lentiviral particles were produced using the Lenti-X Single Shot system (Clontech). Following lentiviral infection, stably transduced pools were generated using puromycin selection (HCT 116: 2 µg/ml, UM-UC-3: 2 µg/ml, UM-UC-5: 3 µg/ml). Sources, STAG2 status and authentication information (STR fingerprinting at Eurofins Genomics, Germany) of cell lines used in this study are provided in *Supplementary file 2*. All cell lines were tested negatively for mycoplasma contamination and have been authenticated by STR fingerprinting. Bladder cancer cell lines J82, RT4 and VM-CUB-3 and Ewing sarcoma cell line SK-N-MC are contained within the ICLAC list of commonly misidentified cell lines but have been STR verified for this study.

## siRNA transfection, cell viability, competition assay and apoptosis assay

For knockdown experiments, cells were transfected with ON-TARGETplus SMARTpool siRNA duplexes (Dharmacon, Lafayette, CO, US) and the Lipofectamine RNAiMAX reagent according to the manufacturer's instructions (Invitrogen, Waltham, MA, US). HCT 116 chromosome spreads, apoptosis assay, immunoblotting, immunofluorescence and live cell imaging experiments were performed using a final siRNA concentration of 20 nM. Cell viability and crystal violet staining assays were performed using 10 nM siRNA. hTERT RPE-1 cells and bladder cancer cells were transfected with 10 nM siRNA for cell viability assays, crystal violet staining and chromosome spreads. The UM-UC-3 FLAG-STAG2 cell line was transfected with 20 nM siRNA for immunoblotting. Ewing sarcoma cell lines were co-transfected with 50 nM ON-TARGETplus SMARTpool siRNAs (Dharmacon, Lafayette, CO, US) plus pRetro-Super (*Brummelkamp et al., 2002*) using Lipofectamin Plus reagent (Invitrogen, Waltham, MA, US). The next day cells were subjected to puromycin (1 µg/ml) selection for 72 hr (*Ban et al., 2014*), and cultured for two additional days. Viability was determined using Cell-Titer-Glo (Promega, Madison, WI, US), and by staining with crystal violet (Sigma-Aldrich, St. Louis, MO, US; HT901). For sgRNA competition assays, Cas9-GFP was expressed constitutively (HCT 116) or was induced by doxycycline addition (KBM-7). mCherry and sgRNAs were introduced by lentiviral transduction. The fraction of mCherry-positive cells was determined at the indicated time points using a Guava easycyte flow cytometer (Merck Millipore, Germany) and normalized to the first measurement and sequentially to control sgRNAs (non-targeting for HCT 116 and STAG2_19 for KBM-7). Apoptosis was analyzed using the IncuCyte Caspase-3/7 Apoptosis Assay (Essen BioScience, Ann Arbour, MI, US).

## Cell extracts for immunoblotting and FLAG-immunoprecipitation

Cell pellets were resuspended in extraction buffer (50 mM Tris Cl pH 8.0, 150 mM NaCl, 1% Nonidet P-40 supplemented with Complete protease inhibitor mix (Roche, Switzerland) and Phosphatase Inhibitor cocktails (Sigma-Aldrich, St. Louis, MO, US; P5726 and P0044)); for *Figure 1—figure supplement 1* (KBM-7) and *Figure 3—figure supplement 1*, pellets were resuspended in (25 mM Tris-HCl pH 7.5, 100 mM NaCl, 5 mM MgCl2, 0.2% NP-40, 10% glycerol, 1 mM NaF, Complete protease inhibitor mix (Roche, Switzerland), Benzonase (VWR, Radnor, PA, US)) and lysed on ice. For *Figure 1—figure supplement 2B* and *Figure 3—figure supplement 3C*, lysates were spun down for 10 min, followed by FLAG-immununoprecipitation using anti-FLAG M2-Agarose Affinity Gel (Sigma-Aldrich, St. Louis, MO, US) for two hours and washing with lysis buffer. Input lysates and immunoprecipitates were resuspended in SDS sample buffer and heated to 95℃.

## Immunofluorescence, live cell imaging and chromosome spreads

For immunofluorescence, cells were fixed with 4% paraformaldehyde for 15 min, permeabilized with 0.2% Triton X-100 in PBS for 10 min and blocked with 3% BSA in PBS containing 0.01% Triton X-100.

Cells were incubated with primary and secondary antibody (Alexa 594, Molecular Probes, Eugene, OR, US), DNA was counterstained with Hoechst 33342 (Molecular Probes, Eugene, OR, US; H3570) and tubulin was sequentially stained with an FITC-conjugated mouse anti-tubulin antibody (Sigma-Aldrich, St. Louis, MO, US; F2168)). Coverslips and chambers were mounted with ProLong Gold (Molecular Probes, Eugene, OR, US). Images were taken with an Axio Plan2/AxioCam microscope and processed with MrC5/Axiovision software (Zeiss, Germany). An IncuCyte (EssenBioScience, Ann Arbor, MI, US) imaging system was used to record live cells, and duration of mitosis was determined by measuring the time from mitotic cell rounding until anaphase onset. Data analysis was performed with Microsoft Excel 2013 and GraphPad Prism 7 (GraphPad Sofware, La Jolla, CA, US). Significance levels were quantified using unpaired t test. For chromosome spread analysis, nocodazole was added to the medium for 60 min at 100 ng/ml. Cells were harvested and hypotonically swollen in 40% medium/60%tap water for 5 min at room temperature. Cells were fixed with freshly made Carnoy's solution (75% methanol, 25% acetic acid), and the fixative was changed three times. For spreading, cells in Carnoy's solution were dropped onto glass slides and dried. Slides were stained with 5% Giemsa (Merck, Germany) for 4 min, washed briefly in tap water and air dried. For chromosome spread analysis two independent slides were scored blindly for each condition.

### Bioinformatic analysis of STAG1 and STAG2 mRNA expression

STAG2 mRNA expression in tumors (TCGA Research Network: http://cancergenome.nih.gov/) and STAG1 and STAG2 expression in normal human tissues (The Genotype-Tissue Expression (GTEx) project, www.gtexportal.org) (*GTEx Consortium, 2013*) were analyzed based on the data available from UCSC Xena version 2016-04-12. Expression levels correspond to transcripts per million (TPM). The downloaded values were log2(TPM +0.001) values which were transformed to TPM values before generating the figures. *STAG2* mutation data were downloaded from cBioPortal (*Cerami et al., 2012*; *Gao et al., 2013*) git repository in Sep. 2016. Samples were labeled as *STAG2* mutated if they carried an alteration resulting in an amino acid change in STAG2 (i.e. missense or nonsense point mutation or small insertion or deletion). Gene deletions or amplifications were not considered.

## Acknowledgements

We would like to thank Monika Kriz and Renate Schnitzer for clonal analysis. The IMP is supported by Boehringer Ingelheim. Research in the laboratory of J-MP is funded by the Austrian Science Fund (SFB-F34 and Wittgenstein award Z196-B20) and the Austrian Research Promotion Agency (Headquarter grants FFG-834223 and FFG-852936, Laura Bassi Centre for Optimized Structural Studies grant FFG-840283). Research in the laboratory of JZ was funded by a Starting Grant of the European Research Council (ERC no. 336860) and SFB grant F4710 of the Austrian Science Fund (FWF). Research on Ewing sarcoma in the laboratory of HK was funded by the Austrian Science Fund ERA-Net grant I 1225-B19. Work in the lab of FXR was funded by a grant from Fundación Científica de la Asociación Española Contra el Cáncer, Madrid, Spain. Research in the laboratory of TW is supported by National Institute of Health grant R01CA169345 and an Innovation Grant from Alex's Lemonade Stand.

## Additional information

### Competing interests

SL: Simone Lieb is a full-time employee of Boehringer Ingelheim RCV. AS: Andreas Schlattl is a full-time employee of Boehringer Ingelheim RCV. MAP: Mark Pearson is a full-time employee of Boehringer Ingelheim RCV. NK: Norbert Kraut is a full-time employee of Boehringer Ingelheim RCV. MP: Mark Petronczki is a full-time employee of Boehringer Ingelheim RCV. The other authors declare that no competing interests exist.

### Funding

| Funder | Grant reference number | Author |
| --- | --- | --- |
| Austrian Science Fund | SFB-F34 | Jan-Michael Peters |

| Austrian Science Fund | Wittgenstein award Z196-B20 | Jan-Michael Peters |
| Österreichische Forschungs-förderungsgesellschaft | FFG-834223 | Jan-Michael Peters |
| Österreichische Forschungs-förderungsgesellschaft | FFG-852936 | Jan-Michael Peters |
| Österreichische Forschungs-förderungsgesellschaft | FFG-840283 | Jan-Michael Peters |
| European Research Council | ERC no. 336860 | Johannes Zuber |
| Austrian Science Fund | SFB grant F4710 | Johannes Zuber |
| Austrian Science Fund | ERA-Net grant I 1225-B19 | Heinrich Kovar |
| Fundación Científica Asociación Española Contra el Cáncer | | Francisco X Real |
| National Institutes of Health | R01CA169345 | Todd Waldman |
| Alex's Lemonade Stand Foundation for Childhood Cancer | Innovation Grant | Todd Waldman |
| Boehringer Ingelheim | | Simone Lieb<br>Andreas Schlattl<br>Mark A Pearson<br>Norbert Kraut<br>Mark Petronczki |

The funders had no role in study design, data collection and interpretation, or the decision to submit the work for publication.

## Author contributions

PvdL, Conceptualization, Resources, Supervision, Funding acquisition, Visualization, Methodology, Writing—original draft, Project administration, Writing—review and editing; SL, Conceptualization, Data curation, Formal analysis, Investigation, Visualization, Methodology, Writing—original draft, Writing—review and editing; JJ, Formal analysis, Investigation, Visualization, Methodology, Writing—review and editing; GW, CPS, KF, AS, RS, TH, Investigation, Methodology; JB, Formal analysis, Investigation, Visualization, Methodology; HK, Conceptualization, Methodology; FXR, JZ, Conceptualization, Resources, Supervision, Funding acquisition, Writing—review and editing; TW, Funding acquisition, Writing—review and editing; MAP, Resources, Funding acquisition, Writing—review and editing; NK, Conceptualization, Resources, Writing—review and editing; J-MP, Resources, Writing—review and editing; MP, Conceptualization, Resources, Supervision, Funding acquisition, Methodology, Project administration, Writing—review and editing

## Author ORCIDs

Julian Jude, http://orcid.org/0000-0002-9091-9867
Raphaela Schwentner, http://orcid.org/0000-0001-6839-0322
Johannes Zuber, http://orcid.org/0000-0001-8810-6835
Mark Petronczki, http://orcid.org/0000-0003-0139-5692

# Additional files

## Supplementary files

• Supplementary file 1. Table of sgRNA sequences used in this study.

• Supplementary file 2. Table of cell lines used in this study. Sources, STAG2 status and authentication information (STR fingerprinting) of cell lines are listed.

## Major datasets

The following previously published datasets were used:

| Author(s) | Year | Dataset title | Dataset URL | Database, license, and accessibility information |
|---|---|---|---|---|
| Cerami E, Gao J, Dogrusoz U, Gross BE, Sumer SO, Aksoy BA, Jacobsen A, Byrne CJ, Heuer ML, Larsson E, Antipin Y, Reva B, Goldberg AP, Sander C, Schultz N | 2012 | The cBio cancer genomics portal: an open platform for exploring multidimensional cancer genomics data | https://github.com/cBio-Portal/datahub/tree/master/public | Publicly available via the cBioPortal GitHub (https://github.com/cBioPortal) |
| Vivian J, Rao AA, Nothaft FA, Ketchum C, Armstrong J, Novak A, Pfeil J, Narkizian J, Deran AD, Musselman-Brown A, Schmidt H, Amstutz P, Craft B, Goldman M, Rosenbloom K, Cline M, O'Connor B, Hanna M, Birger C, Kent WJ, Patterson DA, Joseph AD, Zhu J, Zaranek S, Getz G, Haussler D, Paten B | 2017 | Toil enables reproducible, open source, big biomedical data analyses | https://xenabrowser.net/datapages/?dataset=TcgaTargetGtex_rsem_gene_tpm&host=https://toil.xenahubs.net | Publicly available via the UCSC Xena data hub (http://xena.ucsc.edu/) |

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
