## [Decision Letter]

Thank you for submitting your article "Hardwired synthetic lethality within the cohesin complex in human cancer cells" for consideration by *eLife*. Your article has been favorably evaluated by Sean Morrison (Senior Editor) and three reviewers, one of whom, Andrea Musacchio (Reviewer #1), is a member of our Board of Reviewing Editors.

The reviewers have discussed the reviews with one another and the Reviewing Editor has drafted this decision to help you prepare a revised submission.

Summary:

Van der Lelij et al. reveal a synthetic lethal interaction between the cohesin subunit STAG2 and its paralog STAG1. siRNA-meditated depletion of STAG1 leads to cohesion defects and subsequent cell death in STAG2 knockout cells but hardly if at all in wild type cells. The dependency on STAG1 for viability holds true in STAG2-deficient, but not proficient, bladder and Ewing sarcoma cell lines. Re-expressing STAG2 in one of these cell lines restores viability in the absence of STAG1. Overall, the manuscript presents an important conclusion based on a set of well-performed experiments. The paper should be of interest to the readers of *eLife*, but also to the clinic given that STAG2 is frequently mutated in human cancers.

Essential revisions:

The reviewers suggested a few additions that would strengthen your conclusions. We realize that addressing these may further delay publication of the work, and therefore we interpret these comments as suggestions rather than as strict requirements. We note that you might have already collected data that address at least some of these points, in which case you may elect to include them in the revised manuscript.

1) Reviewer 3 notes that it would be helpful to see the effect of STAG1 depletion on non-transformed cells such as RPE1 or BJ-Tert cells. If these cells are far less sensitive to STAG1 depletion than are STAG2 deficient cancer cells, this would further support the authors point that targeting STAG1 may be a good idea for anti-cancer therapy. Including such experiments could give an idea about the possible toxicity of this approach to non-cancer cells.

2) Reviewer 2 notes that it would be helpful to add a comment on whether STAT1 is substantially upregulated at the mRNA and protein levels in all the cell lines and patient derived cell lines. The reviewer would like to see a comprehensive analysis of this in the supplement, and also comments that this is important as it establishes whether STAG2/1 is dominant in cell types or if the genes encoding these two subunits are co-expressed.

3) Reviewer 2 suggests that the statement: "Thus the genetic interaction between STAG paralogs is context independent and conserved in three major human malignancies…." be revised to mention only the "conserved in three major human malignancies". The reviewer argues that it is unclear whether there are other contexts in which these paralogs do exhibit differential functions (as with paralogs of other complexes, such as SWI/SNF. This comment extends to normal cells). The reviewer notes that there are data emerging in the field that already suggest the differential functions (as shown by differential chromatin occupancy and gene expression upon single paralog loss) for these two paralogs.

4) Reviewer 2 also asks if it would be possible for the authors to compare the gene expression profiles upon loss of STAG1 and loss of other cohesion subunits in isogenic STAG2-/- lines (i.e. the HCT-116 cell lines they made). This would help delineate which other aspects of cohesion function are responsible for specific gene expression programs in this lineage. In this context, the authors propose that "Potential approaches for therapeutic targeting of STAG1 include the inhibition of the interaction between STAG1 and the cohesion ring subunit RAD21…".

---

## [Author Response]

Essential revisions:

The reviewers suggested a few additions that would strengthen your conclusions. We realize that addressing these may further delay publication of the work, and therefore we interpret these comments as suggestions rather than as strict requirements. We note that you might have already collected data that address at least some of these points, in which case you may elect to include them in the revised manuscript.

1) Reviewer 3 notes that it would be helpful to see the effect of STAG1 depletion on non-transformed cells such as RPE1 or BJ-Tert cells. If these cells are far less sensitive to STAG1 depletion than are STAG2 deficient cancer cells, this would further support the authors point that targeting STAG1 may be a good idea for anti-cancer therapy. Including such experiments could give an idea about the possible toxicity of this approach to non-cancer cells.

We thank the reviewer for this suggestion. In the revised version of our manuscript we include a new experiment that shows that depletion of STAG1 in contrast to the depletion of essential cohesin factors does not impact the viability of human telomeraseimmortalized retinal pigment epithelial cells (hTERT RPE-1) (Figure 1—figure supplement 5). The result of this experiment suggests that loss of STAG1 function is tolerated in terms of proliferation in non-cancer cells. This result is also in line with genome-wide CRISPR loss-of-function data (Hart et al., Cell 2015; PMID: 26627737) that show that neither STAG1 nor STAG2 is essential in hTERT RPE-1 cells. We now refer to our new experiment and cite the aforementioned study in the third paragraph of the main text.

2) Reviewer 2 notes that it would be helpful to add a comment on whether STAT1 is substantially upregulated at the mRNA and protein levels in all the cell lines and patient derived cell lines. The reviewer would like to see a comprehensive analysis of this in the supplement, and also comments that this is important as it establishes whether STAG2/1 is dominant in cell types or if the genes encoding these two subunits are co-expressed.

We agree with reviewer 2 that these are important points. In the revised manuscript we provide new experimental data that show that the level of STAG1 protein in the cell models employed in this study is not consistently increased in STAG2-negative cells (Figure 4—figure supplement 1). Furthermore, we provide bioinformatic analyses of 2 STAG1 mRNA expression in pan-cancer tumor samples (n=6621) as well as in bladder cancer tumor samples (n=129) from TCGA. Stratification of samples by STAG2 mutational status reveals that also in patient tumors STAG1 mRNA expression is not elevated in STAG2 mutated versus STAG2 wild-type tumors (new Figure 4—figure supplement 1). Together our experiment and analyses suggests that STAG1 upregulation does not represent a compensatory mechanism required in and employed by cells lacking STAG2 (discussed in the sixth paragraph of the main text).

To address the second point, we compared mRNA expression data of STAG1 and STAG2 in human normal tissues (GTEx dataset). This analysis shows that STAG1 and STAG2 are coexpressed in most if not all tissues analyzed with STAG2 expression levels often surpassing STAG1 expression. The analysis is shown in new Figure 4—figure supplement 2. The fact that STAG2 is expressed in all normal human tissues analyzed supports our conclusion that STAG1 inhibition has the potential to yield a large therapeutic window by sparing normal tissues and selectively affecting STAG2 mutated cancer cells (discussed in the eighth paragraph of the main text).

3) Reviewer 2 suggests that the statement: "Thus the genetic interaction between STAG paralogs is context independent and conserved in three major human malignancies…." be revised to mention only the "conserved in three major human malignancies". The reviewer argues that it is unclear whether there are other contexts in which these paralogs do exhibit differential functions (as with paralogs of other complexes, such as SWI/SNF. This comment extends to normal cells). The reviewer notes that there are data emerging in the field that already suggest the differential functions (as shown by differential chromatin occupancy and gene expression upon single paralog loss) for these two paralogs.

We thank the reviewer for pointing this out. In the revised version of the manuscript, the term “context independent” has been removed from the Abstract and the main text.

4) Reviewer 2 also asks if it would be possible for the authors to compare the gene expression profiles upon loss of STAG1 and loss of other cohesion subunits in isogenic STAG2-/- lines (i.e. the HCT-116 cell lines they made). This would help delineate which other aspects of cohesion function are responsible for specific gene expression programs in this lineage. In this context, the authors propose that "Potential approaches for therapeutic targeting of STAG1 include the inhibition of the interaction between STAG1 and the cohesion ring subunit RAD21…".

The effect of cohesin mutations in cancer on transcriptional programs is a very interesting area that could lie at the heart of the tumor-promoting role of these alterations. Published data reveal that the genetic inactivation of STAG2 in isogenic HCT 116 models causes only very minor changes in gene expression (Solomon et al., Science 2011; PMID: 21852505). As shown in this work, removing STAG1 in addition will result in mitotic arrest and cell death. This cell cycle and viability effect is likely to confound any analysis of gene expression changes. Addressing this question appropriately would require the development of chemical genetic tools to acutely eliminate STAG1 function (e.g.: by induced protein degradation). These tools are currently not available. In the future, the effects of cancer-linked cohesin alterations on gene expression and chromatin organization should be analyzed e.g. in GEMM models of bladder cancer where STAG2 mutations have a high prevalence. Defining the transcriptional deregulation that might underlie the tumor promoting function of cohesin mutations is a key area of future 3 research but in our opinion beyond the scope of this Short Report. Our current study focuses on the synthetic lethality between cohesin subunits that is caused by a loss of sister chromatid cohesion and subsequent mitotic failure.